# Changes in Cartilage Thickness and Denuded Bone Area after Knee Joint Distraction and High Tibial Osteotomy—Post-Hoc Analyses of Two Randomized Controlled Trials

**DOI:** 10.3390/jcm10020368

**Published:** 2021-01-19

**Authors:** Mylène P. Jansen, Susanne Maschek, Ronald J. van Heerwaarden, Simon C. Mastbergen, Wolfgang Wirth, Floris P. J. G. Lafeber, Felix Eckstein

**Affiliations:** 1Department of Rheumatology & Clinical Immunology, University Medical Center Utrecht, Utrecht University, Heidelberglaan 100 (G02.228), 3584CX Utrecht, The Netherlands; S.Mastbergen@umcutrecht.nl (S.C.M.); F.Lafeber@umcutrecht.nl (F.P.J.G.L.); 2Department of Imaging & Functional Musculoskeletal Research, Institute of Anatomy & Cell Biology, Paracelsus Medical University Salzburg & Nuremberg, 5020 Salzburg, Austria; maschek@chondrometrics.de (S.M.); wirth@chondrometrics.de (W.W.); eckstein@chondrometrics.de (F.E.); 3Chondrometrics GmbH, 83395 Freilassing, Germany; 4Centre for Deformity Correction and Joint Preserving Surgery, Kliniek ViaSana, 5451AA Mill, The Netherlands; vanheerwaarden@yahoo.com

**Keywords:** knee joint distraction, high tibial osteotomy, MRI, cartilage, joint-preserving

## Abstract

High tibial osteotomy (HTO) and knee joint distraction (KJD) are joint-preserving treatments that unload the more affected compartment (MAC) in knee osteoarthritis. This post-hoc study compares two-year cartilage-thickness changes after treatment with KJD vs. HTO, and identifies factors predicting cartilage restoration. Patients indicated for HTO were randomized to KJD (KJD_HTO_) or HTO treatment. Patients indicated for total knee arthroplasty received KJD (KJD_TKA_). Outcomes were the MRI mean MAC cartilage thickness and percentage of denuded bone area (dABp) change two years after treatment, using radiographic joint space width (JSW) as the reference. Cohen’s d was used for between-group effect sizes. Post-treatment, KJD_HTO_ patients (*n* = 18) did not show significant changes. HTO patients (*n* = 33) displayed a decrease in MAC cartilage thickness and an increase in dABp, but an increase in JSW. KJD_TKA_ (*n* = 18) showed an increase in MAC cartilage thickness and JSW, and a decrease in dABp. Osteoarthritis severity was the strongest predictor of cartilage restoration. Kellgren–Lawrence grade ≥3 showed significant restoration (*p* < 0.01) after KJD; grade ≤2 did not. Effect sizes between severe KJD and HTO patients were large for MAC MRI cartilage thickness (d = 1.09; *p* = 0.005) and dABp (d = 1.13; *p* = 0.003), but not radiographic JSW (d = 0.28; *p* = 0.521). This suggests that in knee osteoarthritis patients with high disease severity, KJD may be more efficient in restoring cartilage thickness.

## 1. Introduction

Knee osteoarthritis (OA) is the most prevalent form of OA and one of the most common causes of disability worldwide [1]. It poses a major global burden, and is anticipated to increase in the future [2,3]. End-stage knee OA is frequently treated with total knee arthroplasty (TKA), a generally effective and safe treatment [4,5]. However, in younger and more active patients, it involves a risk of failure, and future revision surgery. In these cases, a joint-preserving alternative may be a desired option [6].

In the case of predominantly unicompartmental knee OA, unicompartmental knee arthroplasty (UKA), high tibial osteotomy (HTO), and knee joint distraction (KJD) may be considered as (partly) joint-preserving treatment options [7,8,9,10,11]. As opposed to UKA, only HTO and KJD fully preserve the native joint tissue. HTO permanently unloads the more affected compartment (MAC) of the tibio-femoral joint by (over-)correcting the leg axis. This puts more load on the less affected compartment (LAC), and has shown good long-term survival [12,13]. Further, HTO treatment has shown an increase in radiographic joint space width (JSW) and, in some cases, even cartilage restoration [14,15,16]. However, comparison of JSW before and after HTO may be unreliable, as pseudo-widening of the unloaded compartment may occur due to the induced change in leg axis. Therefore, a direct measurement of cartilage structure is required to evaluate whether HTO has indeed a positive effect on maintenance of cartilage tissue.

KJD has been used for uni- and bi-compartmental knee OA. KJD aims to promote cartilage restoration by temporarily unloading both compartments, using an external fixation frame. KJD also has shown good long-term survival and both radiographic JSW increase and cartilage thickness restoration by MRI [14,17,18,19,20,21,22,23,24].

In a previous randomized controlled trial (RCT) that compared HTO with KJD, the clinical effects (based on patient-reported outcomes) and structural effects (based on radiographic measurements) of KJD and HTO were shown to be similar in patients indicated for HTO, with a leg axis deviation of < 10° [14,21,22]. However, for the reasons provided above, direct cartilage thickness measurements need to be compared between KJD and HTO in order to accurately evaluate the efficacy of both treatment options on cartilage structure. Therefore, the main goal of this study was to compare two-year changes in MRI cartilage thickness and denuded joint surface areas during treatment with KJD vs. HTO. We hypothesized that KJD is more effective in restoring cartilage in the MAC, while avoiding negative effects (more cartilage thinning) on the LAC. The secondary goal was to identify (baseline) factors that can predict cartilage restoration activity as measured on MRI, in order to help select the appropriate patients for that type of therapy.

## 2. Patients and Methods

### 2.1. Patients

Patients were included from two different RCTs: one comparing KJD with HTO, and one comparing KJD with TKA. In the KJD vs. HTO trial, patients with medial compartmental knee OA considered for HTO according to regular clinical practice were included to be treated with either KJD (*n* = 23; KJD_HTO_) or HTO (*n* = 46; HTO) [22]. In the KJD vs. TKA trial, knee OA patients considered for TKA according to regular clinical practice were included for treatment with KJD (*n* = 20; KJD_TKA_) or TKA (*n* = 40; TKA). The TKA patients were not included in this study, because they no longer had their native knee after surgery, and no structural parameters could be analyzed [21]. Inclusion and exclusion criteria for both trials have been previously published, including the following inclusion criteria: radiological joint damage (Kellgren–Lawrence grade > 2 as judged by the orthopedic surgeon), age < 65 years (a TKA placed < 65 years brings an increased revision risk [6]), ability to undergo MRI examination, <10° knee malalignment (preferably treated by realignment surgery), BMI < 35 (mechanical limitations of the distraction frame), and no joint prosthesis elsewhere in the body (because risk of infection in case pin-tract infection occurs) [25]. As part of the inclusion process, in the KJD vs. HTO trial, standing whole-leg radiographs were performed to measure the pre-operative leg axis, while in the KJD vs. TKA trial, these radiographs were performed only in around half of the patients.

Both trials were granted ethical approval by the medical ethical review committee of the University Medical Center Utrecht (protocol numbers 10/359/E and 11/072), registered in the Netherlands Trial Register (trial numbers NL2761 and NL2680), and were performed in accordance with the ethical principles of the Declaration of Helsinki. All patients gave their written informed consent. All actions described in this manuscript were part of the original research protocol and ethical approval as well as patient informed consent; no additional actions were performed in the included patients for these post-hoc analyses.

### 2.2. Treatment

Distraction surgery (KJD) was performed using an external fixation frame (Monotube Triax, Stryker, Kalamazoo, MI, United States) consisting of two dynamic monotubes, bridging the knee medially and laterally and fixated to the tibia and femur using eight half-pins [25]. During surgery, the knee was distracted 2 mm, extended for an additional 1 mm per day during a short hospitalization until 5 mm distraction was reached, confirmed radiographically. Subsequently, patients were discharged with a prophylactic anticoagulant prescribed for use during treatment, and were allowed full weight-bearing on the distracted knee, supported by crutches if needed. After six weeks, the frame and pins were surgically removed during day treatment.

For HTO patients, bi-plane medial-based opening-wedge osteotomy was performed. The method of Miniaci was used to preoperatively define the amount of correction needed, and TomoFix medial high tibial plates and screws (DePuy Synthes, Switzerland) or a Synthes locking compression plate (LCP) system (DePuy Synthes, Switzerland) were used for fixation [26]. After surgery, partial weight-bearing (maximum 20 kg) was allowed for six weeks, after which full weight-bearing was started gradually. Prophylactic anticoagulant was used for six weeks. At 18 months after surgery, the metal plate and screws were removed, to allow imaging at two years.

### 2.3. Image Acquisition and Analysis

In our study, 1.5 T or 3 T MRIs with 3D spoiled gradient recalled imaging sequence with fat suppression (SPGR-fs) were acquired at baseline (before treatment) and two years after treatment. While the MRI field strength differed, as some patients were included in an extended imaging study, the protocols used were the same for both the 1.5 T and 3 T MRI scans. To prevent bias, only patients who underwent MRI scans of sufficient quality to allow analysis and were scanned with the same hardware (1.5 or 3 T) at both baseline and two-year follow-up were included in the analyses. The reasons for insufficient quality to allow analysis were severe motion artifacts or insufficient positioning (e.g., relevant parts of the joint cut-off). There were no constraints regarding concomitant treatment during the two years of follow-up. Cartilage structure in the knee was measured using Chondrometrics Works 3.0 software [27]. The primary and secondary outcome parameters for the present study were the two-year change in mean cartilage thickness over the total subchondral bone area (ThCtAB) of the MAC and the percentage of denuded subchondral bone area (dABp) of the MAC, respectively [28]. On an exploratory basis, longitudinal changes were determined for 16 femorotibial subregions: the central, internal, external, anterior, and posterior tibia; and the central, internal, and external femur for both the MAC and the LAC [27]. Further, location-independent analysis was used to determine the total (summed) thinning and thickening scores across all subregions [29].

Standardized semi-flexed weight-bearing radiographs were performed at the same time points, according to the Buckland–Wright protocol, using an aluminum step wedge as a reference standard for image analysis [30,31]. Using knee-image digital-analysis (KIDA) software, the mean JSW of the MAC was determined [32].

Both MRI and radiograph analyses were performed by experienced observers blinded to the type of intervention and acquisition order. For the radiograph analyses, one observer analyzed all images. For the MRI analyses, two different observers analyzed the images; each of the observers processed pairs of baseline and follow-up of each patient in the same session. Also, the number of patients from each treatment was equally divided between the two observers. The reproducibility of both types of analysis have been reported before in detail [27,32,33,34].

### 2.4. Statistical Analysis

This study is a post-hoc analysis on the data of the original RCTs. Potential differences in baseline characteristics between the three groups (KJD_TKA_, KJD_HTO_, and HTO) were analyzed using one-way ANOVA with, in case of statistically significant differences, post-hoc Tukey HSD tests. In the case of continuous variables that were not normally distributed, Kruskal–Wallis tests, with post-hoc Dunn tests in case of statistically significant differences, were used. For categorical variables, chi-square tests were used.

Changes between pre- and two years post-treatment values for all cartilage thickness and JSW parameters were calculated using paired t-tests. Linear regression was used for comparisons in cartilage thickness and JSW changes over two years between groups, correcting for any significantly different baseline patient characteristics. Consistency was tested by including and excluding baseline cartilage thickness and JSW as confounder. The influence of baseline characteristics on the change in MAC cartilage thickness was tested using linear regression. As leg axis measurements were only available for around half of the KJD_TKA_ patients, this parameter was not used in linear regression models (except when specifically mentioned when testing the effect of leg axis). KJD and HTO patients were divided in subgroups based on the strongest predictor of MAC cartilage thickness change; the same statistical tests for changes over time and differences between groups were applied on these subgroups. Sensitivity analyses were performed by adding the trial in which patients were originally included as a potential confounder. Absolute values are presented with mean ± standard deviation (SD) while changes over time are presented as mean change and 95% confidence interval (95%CI). A *p*-value of <0.05 was considered statistically significant. As a measure of the effect size of the primary and secondary outcome, Cohen’s d was used when comparing changes between different groups. Values of 0.20, 0.50, and 0.80 indicate small, moderate, and large effect sizes, respectively [35].

## 3. Results

### 3.1. Patients

A flowchart of the final patient selection is shown in Figure 1. In the KJD_HTO_ group, one patient was excluded before surgery due to inoperability, two patients received other surgery, and two patients had MRI scans of insufficient quality, leaving 18 patients for analysis. In the HTO group, one patient was excluded before treatment due to anxiety, four patients were lost to follow-up because of comorbidities interfering with follow-up, two patients did not undergo an MRI at two years, two patients had MRI scans with different hardware at baseline and two years, and four patients had MRI scans of insufficient quality, leaving 33 patients for analysis. In the KJD_TKA_ group, one patient received a different surgery within two years of follow-up, and one patient refused imaging at two years, leaving 18 patients for analysis.

The baseline characteristics of the three patient groups are displayed in Table 1. The KJD_TKA_ group had a higher age and Kellgren–Lawrence grade than the HTO and KJD_HTO_ groups. It also had a higher MAC denuded bone area and lower MAC cartilage thickness, pointing towards more severe structural pathology, and a lower leg axis deviation than the other two groups. Between KJD_HTO_ and HTO, there were no statistically significant differences in the baseline characteristics.

For around half of the included patients (46%), 3 T MRIs were performed at baseline and two years (KJD_HTO_ 33%, HTO 52%, KJD_TKA_ 50%); the other patients received 1.5 T MRI scans at baseline and two years. This number was not significantly different statistically between groups (*p* = 0.432), so comparisons between groups were not corrected for field strength.

### 3.2. Longitudinal Changes by Patient Group

MRI MAC cartilage thickness and denuded bone area (Figure 2A,B) in the KJD_HTO_ group showed no significant changes over time; neither did the radiographic MAC JSW (Table 2). The HTO group, in contrast, displayed a significant decrease in MAC cartilage thickness and an increase in MAC denuded bone area. However, the HTO group showed a significant increase in radiographic mean MAC JSW. The KJD_TKA_ group displayed a substantial increase in the MAC cartilage thickness and in mean MAC JSW, and a decrease in MAC denuded bone area.

Differences in cartilage structural change over time between KJD_HTO_ and HTO and between KJD_HTO_ and KJD_TKA_ did not reach statistical significance for any of the three MAC parameters (Table 2). Between KJD_TKA_ and HTO, both the cartilage thickness and denuded bone area showed large, statistically significant differences, while the radiographic JSW did not.

Correcting the between-group comparisons for baseline cartilage thickness, denuded bone area, or JSW values did not change significance for any of the comparisons in all three parameters. The MRI field strength (1.5 T or 3 T) did not have a significant influence on the change in MAC cartilage thickness or denuded bone area in any of the three patient groups (all *p* > 0.3).

In the LAC, most groups did not show a significant change in any of the three parameters (cartilage thickness, denuded bone area, and JSW), except for the KJD_HTO_ group, which showed a significant (but small) increase in LAC denude bone area (Table 2).

Subregional cartilage thickness changes and location-independent cartilage thinning and thickening scores are shown in the supplementary data and supplementary Appendix A.

### 3.3. Prediction of Cartilage Thickness Changes

Kellgren–Lawrence grade and treatment were the strongest statistically significant predictors of MAC cartilage thickness change. A higher Kellgren–Lawrence grade was associated with a greater increase in cartilage thickness during treatment (B = 0.174; R^2^ = 0.255; *p* = 0.002). Detailed results can be found in the supplementary data.

A Kruskal–Wallis test confirmed that, for KJD patients, the distribution of the MAC cartilage thickness change was not the same across the different Kellgren–Lawrence grades (*p* = 0.009). Post-hoc tests identified statistically significant differences between grade 1–3, grade 1–4, grade 2–3, and grade 2–4. Separating KJD patients in those with mild OA (Kellgren–Lawrence grade ≤2; KJD_mild_) and with severe knee OA (Kellgren–Lawrence grade ≥3; KJD_severe_) resulted in the best fit of the univariable regression model (B = 0.209; R^2^ = 0.317; *p* < 0.001): KJD patients with severe OA showed a significantly greater increase in cartilage thickness than those with mild OA.

### 3.4. Longitudinal Changes by Baseline Severity

Since baseline OA severity was the strongest predictor for cartilage tissue changes over time, and had a stronger influence than the trial in which patients were included, the main MRI and JSW outcome parameters are presented here, comparing mild and severe OA groups irrespective of the trial the patients originated from. Additionally, severe KJD patients were compared with severe HTO patients, and mild KJD patients with mild HTO patients. The changes over time for the different groups are shown in Table 3; the differences between the groups are shown in Table 4.

The only patient characteristic significantly different between KJD_mild_ and KJD_severe_ was gender (*p* = 0.017). Corrected for gender, there was a large and statistically significant difference in MAC cartilage thickness change between KJD patients with mild compared to more severe OA: KJD_mild_ patients showed a significant decrease, and KJD_severe_ patients showed a significant increase in cartilage thickness (Figure 3A; Table 3 and Table 4). Similarly, the change in MAC denuded bone area showed a large significant difference between both groups, with only KJD_severe_ patients displaying a significant decrease in denuded bone area over time (Figure 3B; Table 3 and Table 4). The difference in cartilage-structure changes between groups was not as clearly observed by MAC JSW change (Figure 3C; Table 4). Finally, only KJD_mild_ showed a significant negative change with the LAC denuded bone area increasing over time (+0.64; 0.08–1.18; *p* = 0.028). Not correcting for gender did not change significance for any comparison between KJD_mild_ and KJD_severe_.

When dividing HTO patients into HTO_mild_ and HTO_severe_, there were no statistically significant differences in baseline characteristics or in two-year changes between these two groups for any of the MAC MRI or JSW parameters (Table 4).

KJD_severe_ showed a significantly greater cartilage restoration response in the MAC than HTO_severe_, with large effect sizes for both MAC cartilage thickness and denuded bone area (Table 3). This was not observed with MAC JSW change. These comparisons were corrected for age, which was significantly different between the two groups (*p* = 0.009). The changes in all three parameters did not differ significantly between KJD_mild_ and HTO_mild_ (Table 4); there were no statistically significant differences between the two groups (all *p* > 0.05).

Sensitivity analyses, which corrected for the fact that patients in almost all comparisons were included in different trials, are shown in supplementary Appendix A. Correcting for the trial did not change significance for the primary outcome (change in MAC cartilage thickness) or for MAC JSW change. For the change in denuded bone area, the difference was no longer statistically significantly different for any comparison when correcting for the original trial patients in which were included.

## 4. Discussion

The main goal of this study was to compare two-year quantitative cartilage changes during treatment with KJD vs. HTO; we hypothesized that KJD is more effective in restoring cartilage in the MAC, while avoiding negative effects on the LAC. The secondary goal was to identify factors that can predict cartilage restoration activity. Baseline OA severity was the strongest indicator of cartilage restoration response after treatment and independent of treatment, and only severe knee OA patients showed statistically significant cartilage restoration after treatment in both cartilage thickness and denuded bone area, in accordance with radiographic JSW results used as a reference. Contrarily, HTO treated patients showed statistically significant cartilage loss on MRI, while the radiographic JSW of the MAC increased. Patients that received KJD in case of HTO indication, with relatively mild OA as compared to KJD in case of TKA indication, demonstrated no differences in cartilage restoration when compared to HTO-treated patients. Effect sizes were moderate to large, and the changes, although seemingly small in mm and percentage of area, seem to be clinically significant as compared to natural progression of loss in cartilage thickness and increase in denuded bone area in comparable untreated knee OA patients [19]. Discussion of the subregional results can be found in the supplementary data [36,37].

The leg axis deviation, the main reason to indicate a knee OA patient for treatment with HTO, did not have a significant influence on the amount of cartilage restoration (supplementary data). Instead, along with the Kellgren–Lawrence grade, a higher patient age and lower baseline cartilage thickness were the strongest indicators for greater cartilage restoration, likely because both these parameters are significantly associated with a higher Kellgren–Lawrence grade (one-way ANOVA: both *p* < 0.045; data not shown). A Kellgren–Lawrence grading providing compartment-specific instead of knee-specific scores was applied, and there was only one KJD patient whose LAC was scored with a Kellgren–Lawrence grade >2. This patient displayed a relatively large increase in cartilage thickness in the LAC that was comparable to that in the MAC (MAC +0.22 mm; LAC +0.24 mm).

Treatment with HTO demonstrated both cartilage thickness loss and an increase in denuded bone area by MRI of the MAC, whereas radiographic MAC JSW increased. It is therefore likely that the MAC JSW increase is predominantly a result of pseudowidening, due to a mechanical-axis shift that is induced with HTO treatment and not due to actual cartilage thickness gain. Despite the increased loading on the LAC, it did not show significant changes in cartilage thickness. Any changes in cartilage thickness after HTO may therefore be solely natural progression, which means HTO treatment does not (quantitatively) affect the tibiofemoral cartilage two years after treatment. The axis shift appeared to offer enough relief in itself, as these same HTO patients had previously shown a significant increase in clinical patient-reported outcomes over the two-year period after treatment [14,21,22].

The results for HTO patients contradict those found in literature, in which quantitative cartilage restoration was seen, although most studies used second-look arthroscopy to visually score cartilage restoration after HTO, and discrepancies may be expected between such different scoring/measurement methods [38]. Two studies have suggested that cartilage gain occurs after HTO using the same MRI measurement method as used in the current study, although in both cases the increases were not statistically significant [39,40]. Those studies measured the cartilage thickness one year after treatment, instead of after two years. In our study, MRI measurements were not performed at one year after treatment for HTO patients, as the metal plate and screws were still present in the knee. A previous study demonstrated that while two-year values were significantly improved compared to baseline, even better results were seen one year after treatment [18,19]. This observation suggests that the cartilage restoration in the current study could be an underestimation of the true initial restorative effect caused by KJD treatment. Similarly, it could be that the HTO patients would have shown a slight increase in cartilage thickness after one year, but a subsequent loss of cartilage (natural progression) in the second year caused an overall negative two-year effect. This also brings forward a limitation of our study, in that it did not include a natural progression group with similar baseline characteristics as the HTO- and TKA-indicated patients. The question is whether such a population exists for TKA-indicated patients, because indication for TKA needs a past of ineffective conservative treatment. This may make it unethical to keep these patients on conservative treatment for an additional two years to serve as a control group.

Despite severe vs. mild OA being the strongest predictor of cartilage thickness changes after KJD, it should be noted that the R^2^ value of this model was only 0.32, so only 32% of the group variance could be explained by the baseline OA severity. This might be the result of the small number of patients included in the analysis, so it would be worthwhile to perform these analyses in more KJD patients in the future. However, despite the small patient number, the between-group effect sizes for almost all comparisons were moderate to large when dividing groups by severity. This could mean that there are other important factors involved that determine the amount of cartilage restoration that were not included in this study, such as baseline cartilage quality measurements or metabolic joint condition reflected in, for example, synovial fluid marker levels. The fact that the significance of difference in denuded bone area changes between groups changed when correcting for the trial in which patients were treated indicated that there are indeed parameters of importance that were not considered in this trial. These are likely to be structural parameters, since the influence of baseline clinical outcome on cartilage thickness change was found not to be significant (baseline VAS pain, EQ5D, ICOAP, WOMAC, and KOOS, including all subscales, all *p* > 0.1). Future studies that include more parameters, using for example qualitative MRI scans, could provide a better insight into which factors determine cartilage restoration response, and with that might improve the patient selection process.

In conclusion, for patients included in the same trial (KJD vs. HTO), the two treatments showed similar results in MAC cartilage restoration. It was expected that HTO would show worse results in the LAC, but this was not the case. Based on subgroup analyses, it was shown that in patients with severe knee OA, KJD may be more efficient in restoring cartilage thickness than HTO is. In patients with mild knee OA, neither HTO nor KJD treatment results in significant cartilage restoration over two years, and both treatments showed a slight deterioration that is likely the result of natural OA progression. There were no differences between the treatments for changes in the LAC. Based on these results, our research suggests that knee joint distraction as joint-preserving surgery could be a good option in the case of knee OA patients with more severe structural damage. This should be confirmed in a larger trial specifically designed for this purpose.

## Figures and Tables

**Figure 1 jcm-10-00368-f001:**
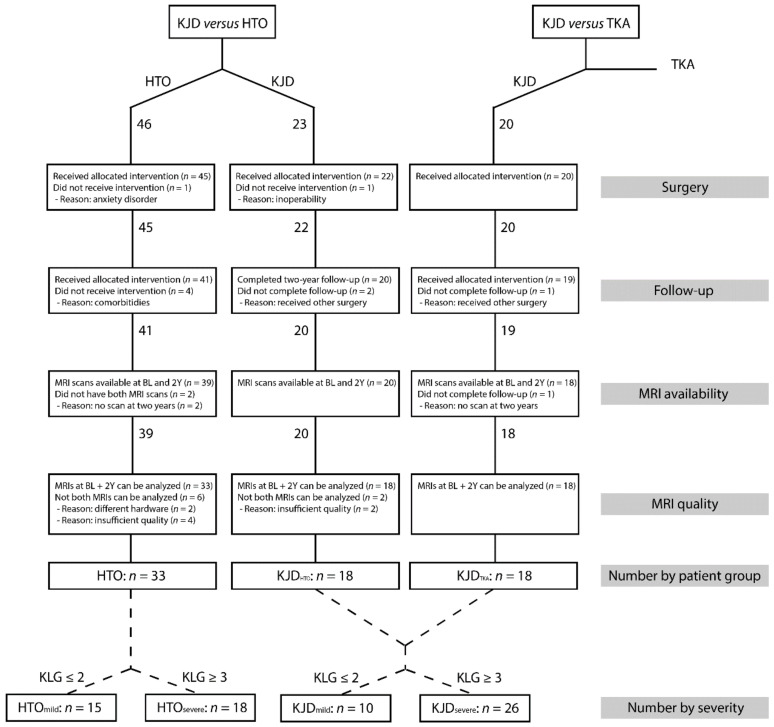
Flowchart of the final patient selection.

**Figure 2 jcm-10-00368-f002:**
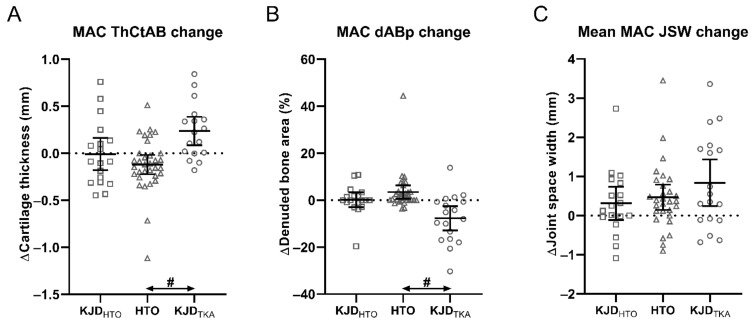
Two-year change in radiographic and MRI cartilage parameters. (**A**) Change in MRI mean cartilage thickness over the total subchondral bone area (ThCtAB) of the more affected compartment (MAC) for the three groups: KJD_HTO_ = KJD patients with indication high tibial osteotomy (HTO) (squares); HTO = HTO patients (triangles); KJD_TKA_ = knee joint distraction (KJD) patients with indication of total knee arthroplasty (circles). (**B**) Change in MRI percentage of denuded subchondral bone area (dABp) of the MAC for the three groups. (**C**) Change in mean radiographic joint space width (JSW) of the MAC for the three groups. Markers represent individual patients, and dashes represent the group mean and 95% confidence interval. Hashes (#) between groups indicate statistically significant differences between each two groups (*p* < 0.05), corrected for baseline patient characteristics significantly different statistically between each two groups.

**Figure 3 jcm-10-00368-f003:**
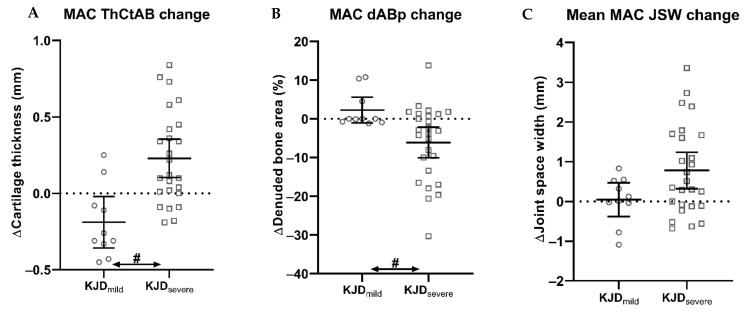
Two-year change in radiographic and MRI cartilage thickness parameters for knee joint distraction (KJD) patients with mild (KJD_mild_, circles) and severe (KJD_severe_, squares) osteoarthritis. (**A**) Change in MRI mean cartilage thickness over the total subchondral bone area (ThCtAB)of the more affected compartment (MAC). (**B**) Change in MRI percentage of denuded subchondral bone area (dABp) of the MAC for both groups. (**C**) Change in mean radiographic joint space width (JSW) of the MAC for both groups. Markers represent individual patients, and dashes represent the group mean and 95% confidence interval. Hashes (#) between groups indicate statistically significant differences between each two groups (*p* < 0.05), corrected for statistically significantly different baseline characteristics.

**Table 1 jcm-10-00368-t001:** Baseline characteristics of the three patient groups.

Parameter	KJD_HTO_ (*n* = 18)	HTO (*n* = 33)	KJD_TKA_ (*n* = 18)	*p*-Value
Age (years), mean (SD)	50.6 (5.3)	49.6 (5.5)	56.6 (6.5)	**<0.001**
Male gender, *n* (%) *	13 (72)	19 (58)	8 (44)	0.240
BMI (kg/m^2^), mean (SD)	27.6 (3.4)	27.3 (3.4)	26.9 (3.8)	0.825
Leg axis (degrees), mean (SD)	5.7 (2.6)	6.1 (2.4)	2.1 (7.0) ^#^	**0.013**
Kellgren–Lawrence grade, *n* (%) *				**0.001**
- Grade 0	0 (0)	1 (3)	0 (0)	
- Grade 1	5 (28)	4 (12)	0 (0)	
- Grade 2	4 (22)	10 (30)	1 (6)	
- Grade 3	9 (50)	14 (42)	7 (39)	
- Grade 4	0 (0)	4 (12)	10 (56)	
Baseline cartilage, mean (SD)				
- MAC ThCtAB (mm)	2.5 (1.0)	2.4 (0.8)	1.8 (0.8)	**0.044**
- LAC ThCtAB (mm)	4.1 (0.8)	4.0 (0.5)	4.0 (1.1)	0.881
- MAC dABp (%)	16 (15)	14 (14)	34 (16)	**<0.001**
- LAC dABp (%)	5.6 (6.6)	3.3 (3.9)	8.4 (12)	0.075
Mean MAC JSW, mean (SD)	2.4 (1.4)	2.2 (1.3)	2.1 (2.2)	0.786

KJD_TKA_ = knee joint distraction (KJD) patients with indication total knee arthroplasty; KJD_HTO_ = KJD patients with indication high tibial osteotomy (HTO); MAC = more affected compartment; LAC = less affected compartment; ThCtAB = mean cartilage thickness over the total subchondral bone area (in mm); dABp = percentage of denuded subchondral bone area; JSW = joint space width. *p*-Values are calculated with one-way ANOVA, with post-hoc Tukey HSD tests in case of statistical significance (bold *p*-values), which showed that all statistically significant differences were between KJD_TKA_ and the other two groups, and there were no statistically significant differences between KJD_HTO_ and HTO. The *p*-values with * were calculated with Chi-square tests; statistically significant differences existed only between KJD_TKA_ and the other two groups. ^#^ Leg axis measurements in the KJD_TKA_ group were available in only eight of 18 patients.

**Table 2 jcm-10-00368-t002:** Two-year changes in the three patient groups.

Parameter	KJD_HTO_	HTO	KJD_TKA_	KJD_HTO_ vs. HTO			KJD_HTO_ vs. KJD_TKA_			KJD_TKA_ vs. HTO		
	Change	Change	Change	Diff	*p*	d	Diff	*p*	d	Diff	*p*	d
MAC ThCtAB (mm)	−0.01 (−0.18–0.16)	−0.12 (−0.22–−0.02)	0.24 (0.08–0.39)	0.11 (−0.07–0.29)	0.225	0.36	−0.01 (−0.29–0.28)	0.954	0.76	−0.35 (−0.58–−0.12)	**0.004**	1.22
LAC ThCtAB (mm)	−0.07 (−0.15–0.00)	−0.02 (−0.09–0.05)	−0.01 (−0.12–0.10)	−0.05 (−0.16–0.06)	0.351	0.28	−0.07 −0.25 –0.11)	0.416	0.34	−0.02 (−0.18–0.14)	0.814	0.06
MAC dABp (%)	0.1 (−3.0–3.3)	3.4 (0.5–6.3)	−7.7 (−12.9–−2.5)	−3.3 (−7.7–1.1)	0.142	0.44	3.6 (−4.3–11.5)	0.366	0.91	11.1 (4.2–18.0)	**0.002**	1.44
LAC dABp (%)	0.5 (0.2–0.8)	0.1 (−0.1–0.3)	0.1 (−1.2–1.4)	0.4 (0.1–0.8)	**0.026**	0.67	0.1 (−1.7–1.9)	0.903	0.23	−0.3 (−1.5–1.0)	0.678	0.02
Mean MAC JSW (mm)	0.31 (−0.11–0.74)	0.47 (0.14–0.79)	0.84 (0.24–1.43)	−0.15 (−0.67–−0.36)	0.547	0.18	−0.02 (−0.97–0.94)	0.969	0.51	−0.36 (−1.16–0.44)	0.374	0.37
Mean LAC JSW (mm)	0.22 (−0.63–1.06)	−0.47 (−1.08–0.13)	0.15 (−0.36–0.66)	0.69 (−0.30–1.68)	0.167	0.42	0.08 (−1.26–1.41)	0.904	0.05	−0.62 (−1.75–−0.50)	0.271	0.44

KJD_TKA_ = knee joint distraction (KJD) patients with indication total knee arthroplasty; KJD_HTO_ = KJD patients with indication high tibial osteotomy (HTO); MAC = most affected compartment; LAC = least affected compartment; ThCtAB = mean cartilage thickness over the total subchondral bone area (in mm); dABp = percentage of denuded subchondral bone area; JSW = joint space width; d = Cohen’s d; Diff = mean difference between groups corrected for statistically significant between-group differences. Changes and differences are shown with mean and 95% confidence interval. Bold *p*-values indicate statistical significance (*p* < 0.05) for changes over time calculated with paired *t*-tests and for differences between groups calculated with linear regression, correcting for statistically significantly different baseline characteristics.

**Table 3 jcm-10-00368-t003:** Two-year changes for mild and severe knee joint distraction and high tibial osteotomy patients.

Parameter	KJD_mild_	KJD_severe_	HTO_mild_	HTO_severe_
MAC ThCtAB (mm)	−0.19 (−0.36–−0.02)	0.23 (0.10–0.35)	−0.10 (−0.20–−0.01)	−0.13 (−0.31–−0.05)
MAC dABp (%)	2.3 (−1.1–5.6)	−6.1 (−10.1–−2.2)	1.2 (−0.2–2.6)	5.3 (0.0–10.5)
Mean MAC JSW	0.04 (−0.38–0.47)	0.78 (0.32–1.24)	0.46 (0.14–0.79)	0.47 (−0.09–1.03)

KJD_mild_ = knee joint distraction (KJD) patients with mild osteoarthritis (OA); KJD_severe_ = KJD patients with severe OA; HTO_mild_ = high tibial osteotomy patients with mild OA; HTO_severe_ = HTO patients with severe OA; MAC = most affected compartment; ThCtAB = mean cartilage thickness over the total subchondral bone area (in mm); dABp = percentage of denuded subchondral bone area; JSW = joint space width; SRM = standardized response mean. Changes are shown with mean and 95% confidence interval.

**Table 4 jcm-10-00368-t004:** Differences in two-year changes between groups based on osteoarthritis severity and treatment.

Parameter	KJD_mild_ vs. KJD_severe_			HTO_mild_ vs. HTO_severe_			KJD_mild_ vs. HTO_mild_			KJD_severe_ vs. HTO_severe_		
	Diff	*p*	d	Diff	*p*	d	Diff	*p*	d	Diff	*p*	d
MAC ThCtAB (mm)	0.46 (0.22–0.70)	**<0.001**	1.47	−0.02 (−0.23–0.19)	0.828	0.08	0.08 (−0.09–0.25)	0.332	0.40	−0.33 (−0.55–−0.11)	**0.005**	1.09
MAC dABp (%)	−8.7 (−16.0–−1.4)	**0.020**	0.97	4.1 (−1.6–9.8)	0.154	0.52	−1.1 (−4.0–1.9)	0.459	0.31	10.5 (3.7–17.2)	**0.003**	1.13
Mean MAC JSW	0.75 (−0.11–1.60)	0.084	0.72	0.01 (−0.66–0.67)	0.985	0.01	0.42 (−0.07–0.91)	0.089	0.75	−0.25 (−1.02–0.52)	0.521	0.28

KJD_mild_ = knee joint distraction (KJD) patients with mild osteoarthritis (OA); KJD_severe_ = KJD patients with severe OA; HTO_mild_ = high tibial osteotomy patients with mild OA; HTO_severe_ = HTO patients with severe OA; MAC = most affected compartment; ThCtAB = mean cartilage thickness over the total subchondral bone area (in mm); dABp = percentage of denuded subchondral bone area; JSW = joint space width; d = Cohen’s d; Diff = mean difference between groups corrected for statistically significant between-group differences. Differences are shown with mean and 95% confidence interval. Bold *p*-values indicate statistical significance (*p* < 0.05), calculated with linear regression, correcting for statistically significantly different baseline characteristics.

## Data Availability

The data presented in this study are available on request from the corresponding author. The data are not publicly available due to ethical restrictions related to participant consent.

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
