# Peer review of "Changes in Cartilage Thickness and Denuded Bone Area after Knee Joint Distraction and High Tibial Osteotomy—Post-Hoc Analyses of Two Randomized Controlled Trials"

_jcm, 2021, doi:10.3390/jcm10020368_

Round 1

Reviewer 1 Report

This is a very original manuscript and, without a doubt, it is about a treatment to take into account in the near future. It is, therefore, an important study for the future performance of the orthopaedic surgeon in degenerative pathology of the knee.

A few minor suggestions that must be resolved:

  1. Authors have to include the reasons why the inclusion criteria have these limits. BMI, malalignment, age and the presence of other kind of prosthesis could not be an excluding factor. If authors think different, it must be justify. Both previous trials had these inclusion criteria and the authors of the present manuscript are some of those who wrote them. That is why this clarification is requested regarding the selection of patients.
  2. Even both trials, on which this manuscript is based, were granted ethical approval by the medical ethical review committee, probably it is mandatory to achieve a new one for this particular study. That is because new actions were performed as the remove plate and screws in all patients of the HTO group. This action could be a limitation in order to achieve a new ethical committee acceptance.
  3. How many observers analyze MRI images? Have you got some intra and interobserver evaluation study? If not, this is something to take into account in the discussion section as an important limitation. Authors have to understand that the observed differences could be attributed to the various interpretations of the different observers. Homogeneity and reliability in the assessment is essential in this type of study.
  4.  

Author Response

We thank the reviewer for the helpful comments by which we feel the manuscript has clearly improved. Below an itemized response on the questions of reviewer 1. Hopefully we answered all questions and addressed all comments to the intention and satisfaction of the reviewer.

  • Reviewer comment:
    Authors have to include the reasons why the inclusion criteria have these limits. BMI, malalignment, age and the presence of other kind of prosthesis could not be an excluding factor. If authors think different, it must be justify. Both previous trials had these inclusion criteria and the authors of the present manuscript are some of those who wrote them. That is why this clarification is requested regarding the selection of patients.
  • Author response:
    As suggested, we have included short explanations for these inclusion criteria, as the reasoning behind them was not previously explained. BMI <35 was a criterion because that was the (weight-bearing) limit for the distraction frame; the malalignment <10 degrees was a criterion because a serious malalignment should be treated with realignment surgery instead; age <65 years was a criterion since it has been shown that in OA patients <65 years, a TKA brings an increased risk of revision surgery (and thus joint-preserving surgery is desired); the presence of a joint prosthesis elsewhere in the body was a criterion because of an increased risk of infection. This has all been included in the revised manuscript [line 111-114 of the revised manuscript with tracked changes].

  • Reviewer comment:
    Even both trials, on which this manuscript is based, were granted ethical approval by the medical ethical review committee, probably it is mandatory to achieve a new one for this particular study. That is because new actions were performed as the remove plate and screws in all patients of the HTO group. This action could be a limitation in order to achieve a new ethical committee acceptance.
  • Author response:
    All actions that were performed and described in this manuscript, including removal of the plates and screws in the HTO group, were included in the original protocol and ethical approval, since two-year follow-up including MRIs were included in the original outcomes of these trials. As such no new ethical approval was required. We have clarified this in the revised manuscript [line 122-124 of the revised manuscript with tracked changes].

  • Reviewer comment:
    How many observers analyze MRI images? Have you got some intra and interobserver evaluation study? If not, this is something to take into account in the discussion section as an important limitation. Authors have to understand that the observed differences could be attributed to the various interpretations of the different observers. Homogeneity and reliability in the assessment is essential in this type of study.
  • Author response:
    This is indeed important, which is why we provided references to studies demonstrating the high reproducibility of both the MRI analyses and the radiograph analyses techniques [line 163-164 of the revised manuscript]. For the radiograph analyses, one observer analyzed all images. For the MRI analyses, two different observers analyzed the images. The number of patients from each treatment was equally divided between the two observers, so they both analyzed a similar number of KJD and HTO patients, but the observers themselves were blinded to the received treatment. Furthermore, each of the readers processed pairs of baseline and follow-up in the same session, and it always was the same reader for baseline and follow-up readings on one study participant, so within-person changes are always measured by the same reader. As such, we have taken multiple measures to ensure that any variability is as limited as possible. We have now described these measures in the revised manuscript as well [line 164-167 of the revised manuscript with tracked changes].

Reviewer 2 Report

I am not sure that the hypothesis can be tested with the chosen methodology

Please consults CONSORT statement for a better reporting of RCTs

Line 107: I can see from the trial registrations that the in/exclusion criteria in the two underlying RCTs are not the same. This is a challenge when trying to fuse the two studies, and needs careful attention in the statistical analyses (adjustment) as well as when interpreting the results.

Line 118: I guess it is not up to the authors to estimate the level of evidence of their own study. Such estimation is biased and adds no value to the manuscript. Please consider to omit.

Lines 119-143: Were there any instructions or constraints regarding concomitant treatment in the 2 year follow-up?

Line 140: What does sufficient quality mean?

Line 142-144: Please clarify which of the outcomes that were primary and which that was secondary. Also add a timing (I assume it is 2 years?). Also, the term ‘primary’ and ‘secondary’ is not complete correct, as I guess that the MRI-outcomes were not the primary outcomes in the underlying RCTs? Please consider adding ‘for the present study’.

Further, I can see from the trial registration that for the KJD vs HTO trial, the MRI cartilage changes determined by decrease in denuded bone areas is the primary outcome (of that trial), yet it has not been reported previously (?). This is quite at odds with current standards of RCT reporting. I recommend that the two studies are reported separately and not fused. I think the patients deserve that their data is used as intended (and as they have been promised).

Title/abstract: This study is a secondary analysis of two RCTs, which should be stated in the title and abstract.

Lines 163-176: The statistical analyses should be adjusted for which of the underlying trials the participants are included from and MRI field strength (1,5 vs 3T) as well as other predefined factors, such as any stratification used in randomisation.

Analyzing within-group changes in RCTs is a mistake and should not be done. RCTs are designed to look for differences between groups. Within group changes are biased and ignores the original study design. Please omit these analyses.

Results: the primary outcome is stated to be the ThCtAB at year 2, but the results present change in ThCtAB. This is not trivial, as the interpretation is different. Please report the ThCtAB at year 2 as intended. This also applies for dABp and JSW.

Figure 2: I guess these are unadjusted observations – please report the adjusted estimates as well, as these better reflect the intended analyses. Consider changing the figures to reflect the values from table 2. The p-values (all of them) hold no information and should be deleted.

Table 2: please remove SRM and within-group p-values. These are not suitable for RCT reporting. Group differences should be reported with mean difference, 95%CI and p-values.

The subgroup analyses are not described in the methods and they come across as very post hoc.

The conclusion does not answer the stated hypothesis by focus mainly on the not-pre-specified subgroup analyses.

Author Response

We really appreciate the detailed feedback of the reviewer on our manuscript and the helpful comments by which we feel the manuscript has clearly improved. Below an itemized response on the questions of reviewer 2. Hopefully we answered all questions and addressed all comments to the intention and satisfaction of the reviewer.

  • Reviewer comment:
    Please consults CONSORT statement for a better reporting of RCTs
  • Author response:
    In response to this suggestion we have reread the CONSORT statement and checklist, and address specific items further in some of the comments below where of relevance. We had largely followed the statement and checklist in the original manuscript, although we purposefully left out some items, since these are post-hoc analyses and the original RCTs have already been covered in multiple publications, where all items from the CONSORT checklist were reported (e.g. reference 14). We refer to these publications in the manuscript where appropriate. However, in line with a later comment below, we agree that it was not clearly indicated that these are post-hoc analyses of RCTs, and have therefore clarified this in the title and abstract of the revised manuscript [line 4-5 and 51 of the revised manuscript with tracked changes].

  • Reviewer comment:
    Line 107: I can see from the trial registrations that the in/exclusion criteria in the two underlying RCTs are not the same. This is a challenge when trying to fuse the two studies, and needs careful attention in the statistical analyses (adjustment) as well as when interpreting the results.
  • Author response:
    There were indeed minor differences in in- and exclusion criteria between the two trials, which were mostly focused on the different treatment indication (either TKA or HTO). In practice, no patients were included in either trial that met an exclusion criterion of the other trial (e.g. no patient was included in the KJD vs TKA trial and had an axis deviation of >10 degrees). Still, this was an important reason why we kept the two KJD groups divided in the first part of the manuscript (to compare KJDHTO with HTO separately, since these were randomized and were included with the same criteria). In the last part of the manuscript patient groups were divided based on OA severity, but we again tested for different baseline characteristics to be able to correct for potential inclusion differences, which is important in interpreting the results as you rightfully comment. This can be found in e.g. line 297-298 of the revised manuscript with tracked changes. However, for some comparisons we did not explicitly mention which characteristics we corrected for, so we have now included this in the revised manuscript [line 324-327].

  • Reviewer comment:
    Line 118: I guess it is not up to the authors to estimate the level of evidence of their own study. Such estimation is biased and adds no value to the manuscript. Please consider to omit.
  • Author response:
    This information was included to make clear that even though these are randomized controlled trials, the level of evidence was not the highest level (I), as RCTs normally would be, but instead a level lower (II), since we combine different RCTs. However, we understand it adds no value to the manuscript, and have removed it from the revised version [line 124-125 of the revised manuscript with tracked changes].

  • Reviewer comment:
    Lines 119-143: Were there any instructions or constraints regarding concomitant treatment in the 2 year follow-up?
  • Author response:
    There were no instructions or constraints regarding concomitant treatment in the two years of follow-up, and in line with this comment we have now included this in the revised manuscript [line 150-151 of the revised manuscript with tracked changes].

  • Reviewer comment:
    Line 140: What does sufficient quality mean?
  • Author response:
    This means sufficient quality to allow reliable image analysis. Reasons for insufficient quality were for example too much motion (motion artefacts) or insufficient image orientation/positioning (e.g. relevant part of the joint cut off, or >3 slices difference in center of the image). To clarify, we have elaborated on this in the revised version of the manuscript [line 148-150 of the revised manuscript with tracked changes].

  • Reviewer comment:
    Line 142-144: Please clarify which of the outcomes that were primary and which that was secondary. Also add a timing (I assume it is 2 years?). Also, the term ‘primary’ and ‘secondary’ is not complete correct, as I guess that the MRI-outcomes were not the primary outcomes in the underlying RCTs? Please consider adding ‘for the present study’.
  • Author response:
    The two-year change in cartilage thickness (ThCtAB) and denuded bone area (dABp) over two years were the primary and secondary outcome, respectively. We did not specify this clearly enough, and indeed these were the primary and secondary outcome for the present study, not for the underlying RCTs. As suggested, we have made these changes in the revised manuscript [line 152-154 of the revised manuscript with tracked changes].

  • Reviewer comment:
    Further, I can see from the trial registration that for the KJD vs HTO trial, the MRI cartilage changes determined by decrease in denuded bone areas is the primary outcome (of that trial), yet it has not been reported previously (?). This is quite at odds with current standards of RCT reporting. I recommend that the two studies are reported separately and not fused. I think the patients deserve that their data is used as intended (and as they have been promised).
  • Author response:
    This is a good point with respect to the original RCTs, we did not realize this before and for both studies the original one- and two-year follow-up analyses have already been published. The informed consent that patients sign, however, allows for any additional (post-hoc) analyses on other outcome parameters; the present study falls under the intended and permitted use of the patients’ data. This has been added line 122-124 of the revised manuscript with track changes

  • Reviewer comment:
    Title/abstract: This study is a secondary analysis of two RCTs, which should be stated in the title and abstract.
  • Author response:
    Indeed it is a post-hoc analysis of two RCTs and we did not make this clear enough. As such, we have made the suggested change to the title and in the abstract [line 4-5 and line 51 of the revised manuscript with tracked changes].

  • Reviewer comment:
    Lines 163-176: The statistical analyses should be adjusted for which of the underlying trials the participants are included from and MRI field strength (1,5 vs 3T) as well as other predefined factors, such as any stratification used in randomisation.
  • Author response:
    To prevent any bias in changes over time, patients with different hardware (1,5T vs 3T) at baseline and follow-up were excluded from these analyses (line 146-148). Also, as we previously mentioned and in line with this comment more clearly highlight in the manuscript [line 223-225 of the revised manuscript with tracked changes], there was no significant difference between groups in the number of patients who had 1,5 vs 3T scans. Furthermore, the field strength did not have a significant influence on the two-year change in primary parameters in any of the patient groups (line 250-252 of the revised manuscript with tracked changes). As described in the manuscript, all statistical tests between groups were corrected for statistically significantly different patient characteristics (line 177-179); since the MRI field strength was not statistically significantly different between groups, we did not correct for it, as we have clarified now specifically in the revised manuscript [line 224-225].

  • Reviewer comment:
    Analyzing within-group changes in RCTs is a mistake and should not be done. RCTs are designed to look for differences between groups. Within group changes are biased and ignores the original study design. Please omit these analyses.
  • Author response:
    As suggested in an earlier comment, we have consulted the CONSORT statement and checklist on reporting RCTs. One of the points in the CONSORT checklist is to report 'for each primary and secondary outcome, results for each group, and the estimated effect size and its precision’. In line with this CONSORT guideline, we consider that the within-group changes should be reported, as well as the between-group changes. We have therefore kept these analyses and estimated effect sizes in the revised manuscript.

  • Reviewer comment:
    Results: the primary outcome is stated to be the ThCtAB at year 2, but the results present change in ThCtAB. This is not trivial, as the interpretation is different. Please report the ThCtAB at year 2 as intended. This also applies for dABp and JSW.
  • Author response:
    We understand we did not report this clearly. The primary outcome should indeed, as we presented in the results, be the two-year change in ThCtAB and not the absolute two-year value. The absolute value at two years can be misleading since it disregards baseline values and as such does not fully reflect the effect of treatment. The same is true for the dABp and JSW, the outcomes here were two-year changes as well. We have made the appropriate changes in the methods section of the revised manuscript [line 152-153 and line 178 of the revised manuscript with tracked changes].

  • Reviewer comment:
    Figure 2: I guess these are unadjusted observations – please report the adjusted estimates as well, as these better reflect the intended analyses. Consider changing the figures to reflect the values from table 2. The p-values (all of them) hold no information and should be deleted.
  • Author response:
    The values in figure 2 fully reflect the MAC ThCtAB, dABp and JSW values from table 2; the mean and 95% confidence interval as seen in the figure and table are identical. The p-values between groups that were included in figure 2 (and table 2) were adjusted for statistically significant baseline differences between each two groups, as described in the methods section and in the legend of the figure (and table). However, following this suggestion, we have removed the p-values from this figure, including those indicating statistical significance of changes within groups as well as those between groups. We replaced the p-values with asterisks and hashes in case of statistical significance within and between groups, respectively, since both should be reported according to CONSORT guidelines. Figure 2 and 3 and corresponding figure legends have been changed in the revised manuscript accordingly [line 233-244 and line 308-318 of the revised manuscript with tracked changes].

  • Reviewer comment:
    Table 2: please remove SRM and within-group p-values. These are not suitable for RCT reporting. Group differences should be reported with mean difference, 95%CI and p-values.
  • Author response:
    As suggested, we have included the group difference with mean difference and 95%CI (p-values were already included, although small mistakes were discovered and corrected here, this did not change any significance). The group differences and 95%CI are corrected for statistically significant differences between groups, as described in the methods and results sections, so that they correspond with the reported p-values. We have made these suggested changes in table 2 and table 4 [line 260-266 and line 291-296 in the revised manuscript with tracked changes]. As mentioned earlier, we consider that reporting the within group changes and effect sizes according to CONSORT guidelines is important and as such have not removed these from the tables.

  • Reviewer comment:
    The subgroup analyses are not described in the methods and they come across as very post hoc.
  • Author response:
    We indeed should have more clearly mentioned this in the methods section. The statistical methods used for the tests were exactly the same as originally described for the KJDTKA, KJDHTO and HTO groups, and as such were covered in the statistical analysis section. However, as suggested, we have included more specifically in the methods section that the division in subgroups based on the strongest predictor of MAC cartilage change was performed, and the same statistical tests were applied as before [line 183-186 of the revised manuscript with tracked changes].

  • Reviewer comment:
    The conclusion does not answer the stated hypothesis by focus mainly on the not-pre-specified subgroup analyses.
  • Author response:
    We think the division based on severity is crucial for our conclusion, partly because the second part of our hypothesis was ‘to identify (baseline) factors that can predict cartilage restoration activity as measured on MRI, in order to help select the appropriate patients for that type of therapy’. For this reason, we focused the conclusion on this, and answered the hypothesis based on the two different severities, to draw a conclusion that is important for patient selection in clinical practice. However, we should indeed have included the part of the hypothesis about the LAC as well, and have now included this in the revised manuscript [line 397-398].

Round 2

Reviewer 2 Report

The authors have not adjusted for participation in the two underlying RCT as suggested. Rather they have tested for differences in baseline characteristics in an attempt to overcome this. However, the purpose of randomisation is to ensure that both known and unknown confounders are equally distributed across groups. While the known possible confounders (i.e. baseline characteristics) may be similar, the unknown confounders remain unknown and not adjusted for. I encourage the authors to adjust for RCT and add as supplementary results (sensitivity analyses), unless the conclusions are significantly different from the main analyses (then they need to be highlighted in the main text).

The authors have re-read the CONSORT statement and argue that one of the points in the CONSORT checklist is to report 'for each primary and secondary outcome, results for each group, and the estimated effect size and its precision’. However, it seems that the authors have misunderstood the CONSORT statement. In the explanation and elaboration document (published in BMJ 2010;340:c869) it is clearly written that: “For each outcome, study results should be reported as a summary of the outcome in each group (for example, the number of participants with or without the event and the denominators, or the mean and standard deviation of measurements), together with the contrast between the groups, known as the effect size.”

The essential feature of a randomised trial is the comparison between groups. Within group analyses do not address a meaningful question: the question is not whether there is a change from baseline, but whether any change is greater in one group than the other. It is not possible to draw valid inferences by comparing SRM from each group. In particular, there is an inflated risk of false positive results. Please omit the tests of within changes (SRMs and p-values).

The conclusion is still overstated with an unwarranted focus on the un-pre-specified subgroup analyses. The conclusion should first state the answer to the research question (HTO vs KJD) and subsequent the subgroup analyses can be mentioned. I disagree that the study results promotes KJD in patients with severe structural damage – at best it is suggestive, but this needs to be confirmed in a large randomised trial specifically designed for this purpose.

The authors state that the second part of the hypothesis was ‘to identify (baseline) factors that can predict cartilage restoration activity as measured on MRI, in order to help select the appropriate patients for that type of therapy’. If this was the case, then why only report radiographic disease severity and not report analyses of all baseline characteristics (age, sex, BMI, leg axis and other characteristics that are not reported here but in other publications such as WOMAC, KOOS etc)?  

Author Response

We thank the reviewer for the additional feedback and suggestions to improve the manuscript, we hope we have addressed all comments to his/her intentions. Below an itemized response on the questions.

  • Reviewer comment:
    The authors have not adjusted for participation in the two underlying RCT as suggested. Rather they have tested for differences in baseline characteristics in an attempt to overcome this. However, the purpose of randomisation is to ensure that both known and unknown confounders are equally distributed across groups. While the known possible confounders (i.e. baseline characteristics) may be similar, the unknown confounders remain unknown and not adjusted for. I encourage the authors to adjust for RCT and add as supplementary results (sensitivity analyses), unless the conclusions are significantly different from the main analyses (then they need to be highlighted in the main text).
  • Author response:
    This is indeed true and we misunderstood this in the previous review round. We agree these sensitivity analyses are good to include, additional to the ones we already did. Correcting for the trial did not change significance for the primary outcome (change in MAC cartilage thickness) or for MAC JSW change. For the change in denuded bone area the difference was no longer statistically significantly different for any comparison. As such, we have included a table with these sensitivity analyses in the supplementary data (supplementary table S1, found under ‘Longitudinal changes by baseline severity: sensitivity analyses’). We refer to this table and the results it shows in the results section [line 331-335 of the revised manuscript with tracked changes] and discussion section [line 399-402] of the revised manuscript.

  • Reviewer comment:
    The authors have re-read the CONSORT statement and argue that one of the points in the CONSORT checklist is to report 'for each primary and secondary outcome, results for each group, and the estimated effect size and its precision’. However, it seems that the authors have misunderstood the CONSORT statement. In the explanation and elaboration document (published in BMJ 2010;340:c869) it is clearly written that: “For each outcome, study results should be reported as a summary of the outcome in each group (for example, the number of participants with or without the event and the denominators, or the mean and standard deviation of measurements), together with the contrast between the groups, known as the effect size.”
    The essential feature of a randomised trial is the comparison between Within group analyses do not address a meaningful question: the question is not whether there is a change from baseline, but whether any change is greater in one group than the other. It is not possible to draw valid inferences by comparing SRM from each group. In particular, there is an inflated risk of false positive results. Please omit the tests of within changes (SRMs and p-values).
  • Author response:
    We agree that SRMs should not be directly compared between groups and displaying them in the same table may give the wrong impression. Therefore, as suggested, we have removed SRMs and p-values from the tables [line 262 and 288 of the revised manuscript with tracked changes] and have now also removed the asterisks from the figures [line 234-235 and line 310-311]. Furthermore, we removed reference to these SRMs and p-values for within-group changes throughout the manuscript.

  • Reviewer comment:
    The conclusion is still overstated with an unwarranted focus on the un-pre-specified subgroup analyses. The conclusion should first state the answer to the research question (HTO vs KJD) and subsequent the subgroup analyses can be mentioned. I disagree that the study results promotes KJD in patients with severe structural damage – at best it is suggestive, but this needs to be confirmed in a large randomised trial specifically designed for this purpose.
  • Author response:
    As suggested, we have included a more specific answer to the research question including only the HTO and KJD patients from the same (HTO vs KJD) trial, before drawing conclusions on the subgroup analyses. We have also toned down the conclusion with regard to promoting KJD in patients with severe structural damage, making clear that the results from the current research suggest that is the case, but that a larger trial specifically designed for this purpose is required to confirm this suggestion [line 408-410 and line 415-417 of the revised manuscript with tracked changes].

  • Reviewer comment:
    The authors state that the second part of the hypothesis was ‘to identify (baseline) factors that can predict cartilage restoration activity as measured on MRI, in order to help select the appropriate patients for that type of therapy’. If this was the case, then why only report radiographic disease severity and not report analyses of all baseline characteristics (age, sex, BMI, leg axis and other characteristics that are not reported here but in other publications such as WOMAC, KOOS etc)?  
  • Author response:
    We did indeed report the analyses of all baseline characteristics; these results can be found in the supplementary data of the original and revised manuscript. We focused on severity, and performed the subgroup analyses based on severity, because it was the strongest predictor and the only predictor left in multivariable regression models. Sex, BMI and leg axis did not have a significant influence, while univariable models showed that patient age did significantly predict cartilage change, but Kellgren-Lawrence grade remained the strongest predictor. All this has been included in the supplementary data. We did not include clinical outcome such as WOMAC and KOOS since in several other publications we have shown that clinical and structural outcome do not correlate well. However, based on this suggestion, we tested the influence of baseline VAS pain, EQ5D, ICOAP, WOMAC and KOOS, and all of their subscales, and none of them had a significant influence on MRI cartilage restoration. We added this in the discussion section [line 402-404 of the revised manuscript with tracked changes].